# Real-valued (Medical) Time Series Generation with Recurrent Conditional GANs

## Abstract

Generative Adversarial Networks (GANs) have shown remarkable success as a framework for training models to produce realistic-looking data. In this work, we propose a Recurrent GAN (RGAN) and Recurrent Conditional GAN (RCGAN) to produce realistic *real-valued multi-dimensional time series*, with an emphasis on their application to medical data. RGANs make use of recurrent neural networks (RNNs) in the generator and the discriminator. In the case of RCGANs, both of these RNNs are conditioned on auxiliary information. We demonstrate our models in a set of toy datasets, where we show visually and quantitatively (using sample likelihood and maximum mean discrepancy) that they can successfully generate realistic time-series. We also describe novel evaluation methods for GANs, where we generate a synthetic labelled training dataset, and evaluate on a *real test set* the performance of a model trained on the *synthetic data*, and vice-versa. We illustrate with these metrics that RCGANs can generate time-series data useful for supervised training, with only minor degradation in performance on *real* test data. This is demonstrated on digit classification from 'serialised' MNIST and by training an early warning system on a medical dataset of 17,000 patients from an intensive care unit. We further discuss and analyse the privacy concerns that may arise when using RCGANs to generate realistic synthetic medical time series data, and demonstrate results from differentially private training of the RCGAN.

## 1 Introduction

Access to data is one of the bottlenecks in the development of machine learning solutions to domain-specific problems. The availability of standard datasets (with associated tasks) has helped to advance the capabilities of learning systems in multiple tasks. However, progress appears to lag in other fields, such as medicine. It is tempting to suggest that tasks in medicine are simply harder - the data more complex, more noisy, the prediction problems less clearly defined. Regardless of this, the dearth of data *accessible* to researchers hinders model comparisons, reproducibility and ultimately scientific progress. However, due to the highly sensitive nature of medical data, its access is typically highly controlled, or require involved and likely imperfect de-identification. The motivation for this work is therefore to exploit and develop the framework of generative adversarial networks (GANs) to generate realistic *synthetic* medical data. This data could be shared and published without privacy concerns, or even used to augment or enrich similar datasets collected in different or smaller cohorts of patients. Moreover, building a system capable of synthesizing realistic medical data implies modelling the processes that generates such information, and therefore it can represent the first step towards developing a new approach for creating predictive systems in medical environments.

Beyond the utility to the machine learning research community, such a tool stands to benefit the medical community for use in training simulators. In this work, we focus on synthesising real-valued time-series data as from an Intensive Care Unit (ICU). In ICUs, doctors have to make snap decisions under time pressure, where they cannot afford to hesitate. It is already standard in medical training to use simulations to train doctors, but these simulations often rely on hand-engineered rules and physical props. Thus, a model capable of generating diverse and realistic ICU situations could have an immediate application, especially when given the ability to condition on underlying 'states' of the patient.

The success of GANs in generating realistic-looking images (Radford et al., 2015; Ledig et al., 2016; Gauthier, 2014; Reed et al., 2016) suggests their applicability for this task, however limited work has exploited them for generating *time-series* data. In addition, evaluation of GANs remains a largely-unsolved problem, with researchers often relying on visual evaluation of generated examples, an approach which is both impractical and inappropriate for multi-dimensional medical time series. For example (Vondrick et al., 2016) present a method to use convolutional GANs specifically designed to generate video sequences, and the results were visually evaluated with Amazon Mechanical Turk. In (Oord et al., 2016a), authors present a method for voice synthesis based on dilated convolutions, which is also evaluated by humans. This voice synthesis model has been very recently improved by introducing an RNN-based network that generates the spectrogram of the signal (Shen et al., 2017).

The primary contributions of this work are:

1. Demonstration of a method to generate multivariate real-valued sequences using adversarial training and recurrent neural networks.

2. Showing novel approaches for evaluating GANs.

3. Generating synthetic medical time series data.

4. Empirical privacy analysis of both GANs and differential private GANs.

## 2 RELATED WORK

Since their inception in 2014 (Goodfellow et al., 2014), the GAN framework has attracted significant attention from the research community, and much of this work has focused on image generation (Radford et al., 2015; Ledig et al., 2016; Gauthier, 2014; Reed et al., 2016). Notably, (Choi et al., 2017) designed a GAN to generate synthetic electronic health record (EHR) datasets. These EHRs contain binary and count variables, such as ICD-9 billing codes, medication, and procedure codes. Their focus on discrete-valued data and generating snapshots of a patient is complementary to our real-valued, time series focus. Future work could combine these approaches to generate multi-modal synthetic medical time-series data.

The majority of sequential data generation with GANs has focused on discrete tokens useful for natural language processing (Yu et al., 2016), where an alternative approach based on Reinforcement Learning (RL) is used to train the GAN. We are aware of only one preliminary work using GANs to generate *continuous-valued* sequences, which aims to produce polyphonic music using a GAN with LSTM generator and discriminator (Mogren, 2016). The primary differences are architectural: we do not use a bidirectional discriminator, and outputs of the generator are not fed back as inputs at the next time step. Moreover, we introduce also a conditional version of this Recurrent GAN.

Conditional GANs (Mirza & Osindero, 2014; Gauthier, 2014) condition the model on additional information and therefore allow us to direct the data generation process. This approach has been mainly used for image generation tasks (Radford et al., 2015; Mirza & Osindero, 2014; Antipov et al., 2017). Recently, Conditional GAN architectures have been also used in natural language processing, including translation (Yang et al., 2017) and dialogue generation (Li et al., 2017), where none of them uses an RNN as the preferred choice for the discriminator and, as previously mentioned, a RL approach is used to train the models due to the discrete nature of the data.

In this work, we also introduce some novel approaches to evaluate GANs, using the capability of the generated synthetic data to train supervised models. In a related fashion, a GAN-based semi-supervised learning approach was introduced in (Salimans et al., 2016). However, our goal is to generate data that can be used to train models for tasks that are unknown at the moment the GAN is trained.

We briefly explore the use of differentially private stochastic gradient descent (Abadi et al., 2016) to produce a RGAN with stronger privacy guarantees, which is especially relevant for sensitive medical data. An alternate method would be to use the PATE approach (Papernot et al., 2016) to train the discriminator. In this case, rather than introducing noise into gradients (as in (Abadi et al., 2016)), a student classifier is trained to predict the noisy votes of an ensemble of teachers, each trained on disjoint sets of the data.

# 3 MODELS: RECURRENT GAN AND RECURRENT CONDITIONAL GAN

The model presented in this work follows the architecture of a regular GAN, where both the generator and the discriminator have been substituted by recurrent neural networks. Therefore, we present a Recurrent GAN (RGAN), which can generate sequences of real-valued data, and a Recurrent Conditional GAN (RCGAN), which can generate sequences of real-value data subject to some conditional inputs. As depicted in Figure 1a, the generator RNN takes a different random seed at each time step, plus an additional input if we want to condition the generated sequence with additional data. In Figure 1b, we show how the discriminator RNN takes the generated sequence, together with an additional input if it is a RCGAN, and produces a classification as synthetic or real for each time step of the input sequence.

Specifically, the discriminator is trained to minimise the average negative cross-entropy between its predictions *per time-step* and the labels of the sequence. If we denote by $\mathrm{RNN}(X)$ the vector or matrix comprising the $T$ outputs from a RNN receiving a sequence of $T$ vectors $\{\mathbf{x}_t\}_{t=1}^T$ ($\mathbf{x}_t \in \mathbb{R}^d$), and by $\mathrm{CE}(\mathbf{a}, \mathbf{b})$ the *average* cross-entropy between sequences $\mathbf{a}$ and $\mathbf{b}$, then the discriminator loss for a pair $\{X_n, \mathbf{y}_n\}$ (with $X_n \in \mathbb{R}^{T \times d}$ and $\mathbf{y}_n \in \{1, 0\}^T$) is:

$$\mathrm{D}_{\mathrm{loss}}(X_n, \mathbf{y}_n) = -\mathrm{CE}(\mathrm{RNN}_{\mathrm{D}}(X_n), \mathbf{y}_n)$$

For real sequences, $\mathbf{y}_n$ is a vector of 1s, or 0s for synthetic sequences. In each training minibatch, the discriminator sees both real and synthetic sequences.

The objective for the generator is then to 'trick' the discriminator into classifying its outputs as true, that is, it wishes to minimise the (average) negative cross-entropy between the discriminator's predictions on *generated* sequences and the 'true' label, the vector of 1s (we write as $\mathbf{1}$);

$$\mathrm{G}_{\mathrm{loss}}(Z_n) = \mathrm{D}_{\mathrm{loss}}(\mathrm{RNN}_{\mathrm{G}}(Z_n), \mathbf{1}) = -\mathrm{CE}(\mathrm{RNN}_{\mathrm{D}}(\mathrm{RNN}_{\mathrm{G}}(Z_n)), \mathbf{1})$$

Here $Z_n$ is a sequence of $T$ points $\{\mathbf{z}_t\}_{t=1}^T$ sampled *independently* from the latent/noise space $\mathbf{Z}$, thus $Z_n \in \mathbb{R}^{T \times m}$ since $\mathbf{Z} = \mathbb{R}^m$. Initial experimentation with non-independent sampling did not indicate any obvious benefit, but would be a topic for further investigation.

In this work, the architecture selected for both discriminator and generator RNNs is the LSTM (Hochreiter & Schmidhuber, 1997).

In the conditional case (RCGAN), the inputs to each RNN are augmented with some conditional information $\mathbf{c}_n$ (for sample $n$, say) by concatenation at *each* time-step;

$$\mathbf{z}_{nt} \to [\mathbf{z}_{nt}; \mathbf{c}_n] \qquad \mathbf{x}_{nt} \to [\mathbf{x}_{nt}; \mathbf{c}_n]$$

In this way the RNN cannot discount the conditional information through forgetting.

Promising research into alternative GAN objectives, such as the Wasserstein GAN (Arjovsky et al., 2017; Gulrajani et al., 2017) unfortunately do not find easy application to RGANs in our experiments. Enforcing the Lipschitz constraint on an RNN is a topic for further research, but may be aided by use of unitary RNNs (Arjovsky et al., 2016; Hyland & Rätsch, 2017).

All models and experiments were implemented in python with scikit-learn (Pedregosa et al., 2011) and Tensorflow (Abadi et al., 2015), and the code is available in a public git repository: ANON.

## 3.1 EVALUATION

Evaluating the performance of a GAN is challenging. As illustrated in (Theis et al., 2015) and (Wu et al., 2016), evaluating likelihoods, with Parzen window estimates (Wu et al., 2016) or otherwise can be deceptive, and the generator and discriminator losses do not readily correspond to 'visual quality'. This nebulous notion of quality is best assessed by a human judge, but it is impractical and costly to do so. In the imaging domain, scores such as the Inception score (Salimans et al., 2016) have been developed to aid in evaluation, and Mechanical Turk exploited to distribute the human labour. However, in the case of real-valued sequential data, is not always easy or even possible to visually evaluate the generated data. For example, the ICU signals with which we work in this paper, could look completely random to a non-medical expert.

Therefore, in this work, we start by demonstrating our model with a number of toy datasets that can be visually evaluated. Next, we use a set of quantifiable methods (description below) that can be used as an indicator of the data quality.

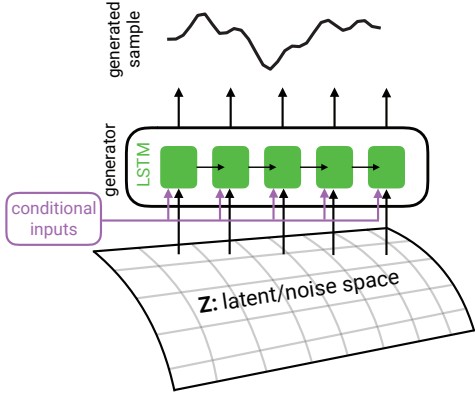

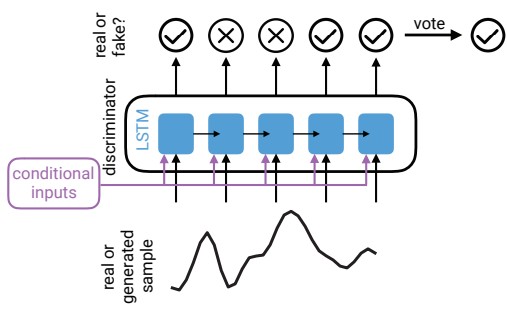

**(a)** The generator RNN takes a different random seed at each temporal input, and produces a synthetic signal. In the case of the RCGAN, it also takes an additional input on each time step that conditions the output.

**(b)** The discriminator RNN takes real/synthetic sequences and produces a classification into real/synthetic for each time step. In the case of the RCGAN, it also takes an additional input on each time step that conditions the output.

**Figure 1:** Architecture of Recurrent GAN and Conditional Recurrent GAN models.

### 3.1.1 MAXIMUM MEAN DISCREPANCY

We consider a GAN successful if it implicitly learns the distribution of the true data. We assess this by studying the samples it generates. This is the ideal setting for maximum mean discrepancy (MMD) (Gretton et al., 2007), and has been used as a training objective for generative moment matching networks (Li et al., 2015). MMD asks if two sets of samples - one from the GAN, and one from the true data distribution, for example - were generated by the same distribution. It does this by comparing *statistics* of the samples. In practice, we consider the squared difference of the statistics between the two sets of samples (the MMD$^2$), and replace inner products between (functions of) the two samples by a kernel. Given a kernel $K : X \times Y \to \mathbb{R}$, and samples $\{x_i\}_{i=1}^N$, $\{y_j\}_{j=1}^M$, an unbiased estimate of MMD$^2$ is:

$$\widehat{\text{MMD}}_u^2 = \frac{1}{n(n-1)} \sum_{i=1}^n \sum_{j \neq i}^n K(x_i, x_j) - \frac{2}{mn} \sum_{i=1}^n \sum_{j=1}^m K(x_i, y_j) + \frac{1}{m(m-1)} \sum_{i=1}^m \sum_{j \neq i}^m K(y_i, y_j)$$

Defining appropriate kernels between time series is an area of active research. However, much of the challenge arises from the need to align time series. In our case, the generated and real samples are already aligned by our fixing of the 'time' axis. We opt then to treat our time series as vectors (or matrices, in the multidimensional case) for comparisons, and use the radial basis function (RBF) kernel using the squared $\ell_2$-norm or Frobenius norm between vectors/matrices; $K(x, y) = \exp(-\|x - y\|^2/(2\sigma^2))$. To select an appropriate kernel bandwidth $\sigma$ we maximise the estimator of the t-statistic of the power of the MMD test between two distributions (Sutherland et al., 2016); $\hat{t} = \frac{\widehat{\text{MMD}}^2}{\sqrt{\hat{V}}}$, where $V$ is the asymptotic variance of the estimator of MMD$^2$. We do this using a split of the validation set during training - the rest of the set is used to calculate the MMD$^2$ using the optimised bandwidth. Following (Sutherland et al., 2016), we define a mixed kernel as a sum of RBF kernels with two different $\sigma$s, which we optimise simultaneously. We find the MMD$^2$ to be more informative than either generator or discriminator loss, and correlates well with quality as assessed by visualising.

### 3.1.2 TRAIN ON SYNTHETIC, TEST ON REAL (TSTR)

We propose a novel method for evaluating the output of a GAN when a supervised task can be defined on the domain of the training data. We call it "**T**rain on **S**ynthetic, **T**est on **R**eal" (TSTR). Simply put, we use a dataset generated by the GAN to train a model, which is then tested on a held-out set of true examples. This requires the generated data to have labels - we can either provide these to a conditional GAN, or use a standard GAN to generate them in addition to the data features. In this work we opted for the former, as we describe below. For using GANs to share synthetic 'de-identified'

data, this evaluation metric is ideal, because it demonstrates the ability of the synthetic data to be used for real applications. We present the pseudocode for this GAN evaluation strategy in Algorithm 1.

---

**Algorithm 1** (TSTR) Train on Synthetic, Test on Real

---

1: `train`, `test` = **split**(`data`)
2: `discriminator`, `generator` = **train_GAN**(`train`)
3: with `labels` from `train`:
4:   `synthetic` = `generator`.**generate_synthetic**(`labels`)
5:   `classifier` = **train_classifier**(`synthetic`, `labels`)
6:   *If validation set available, optionally optimise GAN over classifier performance.*
7: with `labels` and `features` from `test`:
8:   `predictions` = `classifier`.**predict**(`features`)
9:   `TSTR_score` = **score**(`predictions`, `labels`)

---

**Train on Real, Test on Synthetic (TRTS):**   Similar to the TSTR method proposed above, we can consider the reverse case, called "Train on Real, Test on Synthetic" (T**RTS**). In this approach, we use real data to train a supervised model on a set of tasks. Then, we use the RCGAN to generate a synthetic test set for evaluation. In the case (as for MNIST) where the true classifier achieves high accuracy, this serves to act as an evaluation of the RCGAN's ability to generate convincing examples of the labels, and that the features it generates are realistic. Unlike the TSTR setting however, if the GAN suffers mode collapse, TRTS performance will not degrade accordingly, so we consider TSTR the more interesting evaluation.

## 4 LEARNING TO GENERATE REALISTIC SEQUENCES

To demonstrate the model's ability to generate 'realistic-looking' sequences in controlled environments, we consider several experiments on synthetic data. In the experiments that follow, unless otherwise specified, the synthetic data consists of sequences of length 30. We focus on the non-conditional model RGAN in this section.

### 4.1 SINE WAVES

The quality of generated sine waves are easily confirmed by visual inspection, but by varying the amplitudes and frequencies of the real data, we can create a dataset with nonlinear variations. We generate waves with frequencies in $[1.0, 5.0]$, amplitudes in $[0.1, 0.9]$, and random phases between $[-\pi, \pi]$. The left of Figure 2a shows examples of these signals, both real and generated (although they are hard to distinguish).

We found that, despite the absence of constraints to enforce semantics in the latent space (as in (Chen et al., 2016)), we could alter the frequency and phase of generated samples by varying the latent dimensions, although the representation was not 'disentangled', and one dimension of the latent space influenced multiple aspects of the signal.

At this point, we tried to train a recurrent version of the Variational Autoencoder (VAE) (Kingma & Welling, 2013) with the goal of comparing its performance with the RGAN. We tried the implementation proposed in (Fabius & van Amersfoort, 2014), which is arguably the most straightforward solution to implement a Recurrent Variational Autoencoder (RVAE). It consists of replacing the encoder and decoder of a VAE with RNNs, and then using the last hidden state of the encoder RNN as the encoded representation of the input sequence. After performing the reparametrization trick, the resulting encoded representation is used to initialize the hidden state of the decoder RNN. Since in this simple dataset all sequences are of the same length, we also tried an alternative approach in which the encoding of the input sequence is computed as the concatenation of all the hidden states of the encoder RNN. Using these architechtures, we were only capable of generating sine waves with inconsistent amplitudes and frequencies, with a quality clearly inferior than the ones produced by the RGAN. The source code to reproduce these experiments is included in the git repository mentioned before. We believe that this approach needs further research, specially for the task of generating

| | Accuracy |
|------|------------------|
| Real | $0.991 \pm 0.001$ |
| TSTR | $0.975 \pm 0.002$ |
| TRTS | $0.988 \pm 0.005$ |

**Table 1:** Scores obtained by a convolutional neural network when: a) trained and tested on real data, b) trained on synthetic and tested on real data, and c) trained on real and tested on synthetic. In all cases, early stopping and (in the case of the synthetic data) epoch selection were determined using a validation set.

labeled data that will be presented later in this paper, which we also failed to accomplish with the RVAE so far.

## 4.2 SMOOTH FUNCTIONS

Sine waves are simple signals, easily reproduced by the model. In our ultimate medical application, we wish the model to reproduce complex physiological signals which may not follow simple dynamics. We therefore consider the harder task of learning arbitrary smooth signals. Gaussian processes offer a method to sample values of such smooth functions. We use a RBF kernel with to specify a GP with zero-valued mean function. We then draw 30 equally-spaced samples. This amounts to a single draw from a multivariate normal distribution with covariance function given by the RBF kernel evaluated on a grid of equally-spaced points. In doing so, we have specified exactly the probability distribution generated the true data, which enables us to evaluate generated samples under this distribution. The right of Figure 2a shows examples (real and generated) of this experiment. The main feature of the real and generated time series is that they exhibit smoothness with local correlations, and this is rapidly captured by the RGAN.

Because we have access to the data distribution, in Figure 3 we show how the average (log) likelihood of a set of *generated* samples increases under the data distribution during training. This is an imperfect measure, as it is blind to the *diversity* of the generated samples - the oft-observed mode collapse, or 'Helvetica Scenario' (Goodfellow et al., 2014) of GANs - hence we prefer the MMD$^2$ measure (see Figure 3). It is nonetheless encouraging to observe that, although the GAN objective is unaware of the underlying data distribution, the likelihood of the generated samples improves with training.

## 4.3 MNIST AS A TIME SERIES

The MNIST hand-written digit dataset is ubiquitous in machine learning research. Accuracy on MNIST digit classification is high enough to consider the problem 'solved', and generating MNIST digits seems an almost trivial task for traditional GANs. However, generating MNIST sequentially is less commonly done (notable examples are PixelRNN (Oord et al., 2016b), and the serialisation of MNIST in the long-memory RNN literature (Le et al., 2015)). To serialise MNIST, each $28 \times 28$ digit forms a 784-dimensional vector, which is a sequence we can aim to generate with the RGAN. This gives the added benefit of producing samples we can easily assess visually.

To make the task more tractable and to explore the RGAN's ability to generate *multivariate* sequences, we treat each 28x28 image as a sequence of 28, 28-dimensional outputs. We show two types of

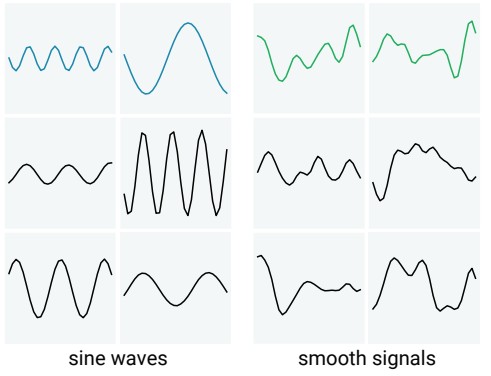

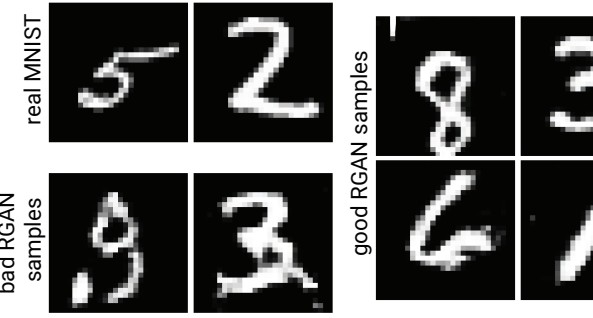

**(a)** Examples of real (coloured, top) and generated (black, lower two lines) samples.

**(b)** Left top: real MNIST digits. Left bottom: unrealistic digits generated at epoch 27. Right: digits with minimal distortion generated at epoch 100.

**Figure 2:** RGAN is capable of generating realistic-looking examples.

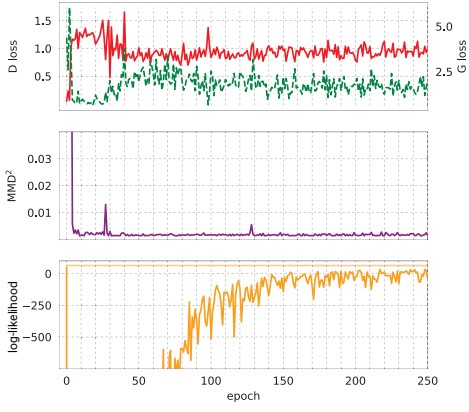

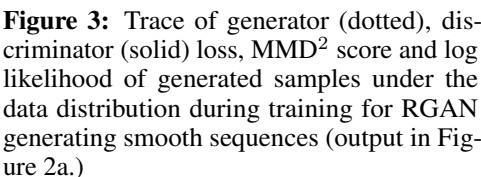

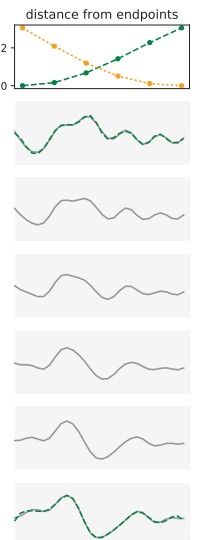

**Figure 4:** Back-projecting training examples into the latent space and linearly interpolating them produces smooth variation in the sample space. Top plot shows sample-space distance from top (green, dashed) sample to bottom (orange, dotted). Distance measure is RBF kernel with bandwidth chosen as median pairwise distance between training samples. The original training examples are shown in dotted lines in the bottom and second-from-top plots.

**Figure 3:** Trace of generator (dotted), discriminator (solid) loss, $MMD^2$ score and log likelihood of generated samples under the data distribution during training for RGAN generating smooth sequences (output in Figure 2a.)

experiment with this dataset. In the first one, we train a RGAN to generate MNIST digits in this sequential manner. Figure 2b demonstrates how realistic the generated digits appear.

For the second experiment, we downsample the MNIST digits to 14x14 pixels, and consider the first three digits (0, 1, and 2). With this data we train a RCGAN and subsequently perform the TSTR (and TRTS) evaluations explained above, for the task of classifying the digits. That is, for the TSTR evaluation, we generate a synthetic dataset using the GAN, using the real training labels as input. We then train a classifier (a convolutional neural network) on this data, and evaluate its performance on the real held-out test set. Conversely, for TRTS we train a classifier on the real data, and evaluate it on a synthetic test dataset generated by the GAN. Results of this experiment are show in Table 1. To obtain error bars on the accuracies reported, we trained the RCGAN five times with different random initialisations. The TSTR result shows that the RCGAN generates synthetic datasets realistic enough to train a classifier which then achieves high performance on real test data. The TRTS result shows that the synthetic examples in the test set match their labels to a high degree, given the accuracy of the classifier trained on real data is very high.

## 5 LEARNING TO GENERATE REALISTIC ICU DATA

One of the main goals of this paper is to build a model capable of generating realistic medical datasets, and specifically ICU data. For this purpose, we based our work on the recently-released Philips eICU database[1]. This dataset was collected by the critical care telehealth program provided by Philips. It contains around 200,000 patients from 208 care units across the US, with a total of 224,026,866 entries divided in 33 tables.

From this data, we focus on generating the four most frequently recorded, regularly-sampled variables measured by bedside monitors: oxygen saturation measured by pulse oximeter (SpO2), heart rate (HR), respiratory rate (RR) and mean arterial pressure (MAP). In the eICU dataset, these variables are measured every five minutes. To reduce the length of the sequences we consider, we downsample to one measurement every fifteen minutes, taking the median value in each window. This greatly speeds up the training of our LSTM-based GAN while still capturing the relevant dynamics of the data (see Figure 6 in the appendix.)

In the following experiments, we consider the *beginning* of the patient's stay in the ICU, considering this a critical time in their care. We focus on the first 4 hours of their stay, which results in 16 measurements of each variable. While medical data is typically fraught with missing values, in this work we circumvented the issue by discarding patients with missing data (after downsampling). After preprocessing the data this way, we end up with a cohort of 17,693 patients. Most restrictive was the requirement for non-missing MAP values, as these measurements are taken invasively.

---

[1]`https://eicu-crd.mit.edu/`

## 5.1 TSTR TASKS IN EICU

The data generated in a ICU is complex, so it is challenging for non-medical experts to spot patterns or trends on it. Thus, one plot showing synthetic ICU data would not provide enough information to evaluate its actual similarity to the real data. Therefore, we evaluate the performance of the ICU RCGAN using the TSTR method.

To perform the TSTR evaluation, we need a supervised task (or tasks) on the data. A relevant question in the ICU is whether or not a patient will become 'critical' in the near future - a kind of early warning system. For a model generating dynamic time-series data, this is especially appropriate, as *trends* in the data are likely most predictive. Based on our four variables (SpO2, HR, RR, MAP) we define 'critical thresholds' and generate binary labels of whether or not that variable will exceed the threshold in the next hour of the patient's stay - that is, between hour 4 and 5, since we consider the first four hours 'observed'. The thresholds are shown in the columns of Table 2a. There is no upper threshold for SpO2, as it is a percentage with 100% denoting ideal conditions. These critical thresholds were selected as follows: starting from rough healthy ranges for adults, we compared with the distributions in our data (ICU patients are not healthy), selected several candidate values for each cutoff, and then ran a grid search over these cutoffs to determine tasks which were sufficiently easy for a random forest (assessed using F1 score). The reasoning here was to ensure that the task would be possible given real data, and to provide a strong baseline against the synthetic data.

As for MNIST, we 'sample' labels by drawing them from the real data labels, and use these as conditioning inputs for the RCGAN. This ensures the label distribution in the synthetic dataset and the real dataset is the same, respecting the fact that the labels are not independent (a patient is unlikely to simultaneously suffer from high and low blood pressure).

Following Algorithm 1, we train the RCGAN for 1000 epochs, saving one version of the dataset every 50 epochs. Afterwards, we evaluate the synthetic data using TSTR. We use cross validation to select the best synthetic dataset based on the classifier performance, but since we assume that it might be also used for unknown tasks, we use only 3 of the 7 tasks of interest to perform this cross validation step (denoted in italics in Table 2a). The results of this experiment are presented in Table 2a, which compares the performance achieved by a random forest classifier that has been trained to predict the 7 tasks of interest, in one experiment with real data and in a different experiment with the synthetically generated data.

## 6 IS THE GAN JUST MEMORISING THE TRAINING DATA?

One explanation for the TSTR performance in MNIST and eICU could be that the GAN is simply "memorising" the training data and reproducing it. If this were the case, then the (potentially private) data used to train the GAN would be leaked, raising privacy concerns when used on sensitive medical data. It is key that the training data for the model should not be recoverable by an adversary. In addition, while the typical GAN objective incentivises the generator to reproduce training examples, we hope that it does not overfit to the training data, and learn an implicit distribution which is peaked at training examples, and negligible elsewhere.

To answer this question we perform three tests - one qualitative, two statistical, outlined in the following subsections. While these evaluations are empirical in nature, we still believe that the proposed and tested privacy evaluation measures can be very useful to quickly check privacy properties of RGAN generated data – but without strong privacy guarantees.

### 6.1 COMPARING THE DISTRIBUTION OF RECONSTRUCTION ERRORS

To test if the generated samples look "too similar" to the training set, we could generate a large number of samples and calculate the distance to the nearest neighbour (in the training set) to each generated sample. We could compare the distribution of these distances with those comparing the generated samples and a held-out test set. However, to get an accurate estimate of the distances, we may need to generate many samples, and correspondingly calculate many pairwise distances. Instead, we *intentionally generate* the nearest neighbour to each training (or test) set point, and then compare the distances.

| | | *SpO2 < 95* | HR < 70 | *HR > 100* |
|---|---|---|---|---|
| AUROC | real | $0.9587 \pm 0.0004$ | $0.9908 \pm 0.0005$ | $0.9919 \pm 0.0002$ |
| | TSTR | $0.88 \pm 0.01$ | $0.96 \pm 0.01$ | $0.95 \pm 0.01$ |
| AUPRC | real | $0.9059 \pm 0.0005$ | $0.9855 \pm 0.0002$ | $0.9778 \pm 0.0002$ |
| | TSTR | $0.66 \pm 0.02$ | $0.90 \pm 0.02$ | $0.84 \pm 0.03$ |
| | random | 0.16 | 0.26 | 0.18 |

| | | *RR < 13* | RR > 20 | MAP < 70 | MAP > 110 |
|---|---|---|---|---|---|
| AUROC | real | $0.9735 \pm 0.0001$ | $0.963 \pm 0.001$ | $0.9717 \pm 0.0001$ | $0.960 \pm 0.001$ |
| | TSTR | $0.86 \pm 0.01$ | $0.84 \pm 0.02$ | $0.875 \pm 0.007$ | $0.87 \pm 0.04$ |
| AUPRC | real | $0.9557 \pm 0.0002$ | $0.891 \pm 0.001$ | $0.9653 \pm 0.0001$ | $0.8629 \pm 0.0007$ |
| | TSTR | $0.73 \pm 0.02$ | $0.50 \pm 0.06$ | $0.82 \pm 0.02$ | $0.42 \pm 0.07$ |
| | random | 0.26 | 0.1 | 0.39 | 0.05 |

**(a)** Performance of random forest classifier for eICU tasks when trained with real data and when trained with synthetic data (test set is real), including random prediction baselines. AUPRC stands for area under the precision-recall curve, and AUROC stands for area under ROC curve.

| | | *SpO2 < 95* | HR < 70 | *HR > 100* |
|---|---|---|---|---|
| AUROC | TSTR (DP) | $0.861 \pm 0.006$ | $0.86 \pm 0.02$ | $0.90 \pm 0.01$ |
| AUPRC | TSTR (DP) | $0.61 \pm 0.01$ | $0.76 \pm 0.04$ | $0.76 \pm 0.03$ |
| | random | 0.16 | 0.27 | 0.16 |

| | | *RR < 13* | RR > 20 | MAP < 70 | MAP > 110 |
|---|---|---|---|---|---|
| AUROC | TSTR (DP) | $0.85 \pm 0.01$ | $0.85 \pm 0.02$ | $0.789 \pm 0.005$ | $0.85 \pm 0.02$ |
| AUPRC | TSTR (DP) | $0.71 \pm 0.02$ | $0.46 \pm 0.03$ | $0.70 \pm 0.01$ | $0.28 \pm 0.03$ |
| | random | 0.26 | 0.09 | 0.39 | 0.05 |

**(b)** Performance of random forest classifier trained on synthetic data generated by differentially private GAN, tested on real data. In each replicate, the GAN was trained with $(\epsilon, \delta)$ differential privacy for $\epsilon = 1$ and $\delta \in [3.55 \times 10^{-12}, 2.26 \times 10^{-9}]$

**Table 2:** TSTR results on eICU tasks using normal (a) and differentially private (b) training. In both cases, epoch from which data is generated was selected using a validation set, considering performance on a subset of the tasks (SpO2 < 95, HR > 100, and RR < 13, denoted in italics). For details on the differentially private setting, see section 7.

We generate these nearest neighbours by minimising the reconstruction error between target $y$ and the generated point; $\mathcal{L}_{\text{recon}(y)}(Z) = 1 - K(G(Z), y)$ where $K$ is the RBF kernel described in Section 3.1.1, with bandwidth $\sigma$ chosen using the median heuristic (Bounliphone et al., 2015). We find $Z$ by minimising the error until approximate convergence (when the gradient norm drops below a threshold).

We can then ask if we can distinguish the *distribution* of reconstruction errors for different input data. Specifically, we ask if we can distinguish the distribution of errors between the training set and the test set. The intuition is that if the model has "memorised" training data, it will achieve identifiably lower reconstruction errors than with the test set. We use the Kolmogorov-Smirnov two-sample test to test if these distributions differ. For the RGAN generating sine waves, the p-value is $0.2 \pm 0.1$, for smooth signals it is $0.09 \pm 0.04$, and for the MNIST experiment shown in Figure 2b it is $0.38 \pm 0.06$. For the MNIST trained with RCGAN (TSTR results in Table 1), the p-value is $0.57 \pm 0.18$. We conclude that the distribution of reconstruction errors is not significantly different between training and test sets in any of these cases, and that the model does not appear to be biased towards reconstructing training set examples.

## 6.2 INTERPOLATION

Suppose that the model has overfit (the implicit distribution is highly peaked in the region of training examples), and most points in latent space map to (or near) training examples. If we take a smooth path in the latent space, we expect that at each point, the corresponding generated sample will have the appearance of the "closest" (in latent space) training example, with little variation until we reach the attractor basin of another training example, at which point the samples switch appearance.

We test this qualitatively as follows: we sample a pair of training examples (we confirm by eye that they don't look "too similar"), and then "back-project" them into the latent space to find the closest corresponding latent point, as described above. We then linearly interpolate between those latent points, and produce samples from the generator at each point. Figure 4 shows an example of this

procedure using the "smooth function" dataset. The samples show a clear incremental variation between start and input sequences, contrary to what we would expect if the model had simply memorised the data.

## 6.3 COMPARING THE GENERATED SAMPLES

Rather than using a nearest-neighbours approach (as in Section 6.1), we can use the MMD three-sample test (Bounliphone et al., 2015) to compare the full set of generated samples. With $X$ being the generated samples, $Y$ and $Z$ being the test and training set respectively, we ask if the MMD between $X$ and $Y$ is less than the MMD between $X$ and $Z$. The test is constructed in this way because we expect that if the model *has* memorised the training data, that the MMD between the synthetic data and the training data will be significantly lower than the MMD between the synthetic data and test data. In this case, the hypothesis that MMD(synthetic, test) $\leq$ MMD(synthetic, train) will be false. We are therefore testing (as in Section 6.1) if our null hypothesis (that the model has *not* memorised the training data) can be rejected. The average p-values we observed were: for the eICU data in Section 5.1: $0.40 \pm 0.05$, for MNIST data in Section 4.3: $0.47 \pm 0.16$, for sine waves: $0.41 \pm 0.07$, for smooth signals: $0.07 \pm 0.04$, and for the higher-resolution MNIST RGAN experiments in Section 4: $0.59 \pm 0.12$ (before correction for multiple hypothesis testing). We conclude that we cannot reject the null hypothesis that the MMD between the synthetic set and test set is at most as large as the MMD between the synthetic set and training set, indicating that the synthetic samples do not look more similar to the training set than they do to the test set.

## 7 TRAINING RGANS WITH DIFFERENTIAL PRIVACY

Although the analyses described in Section 6 indicate that the GAN is not preferentially generating training data points, we are conscious that medical data is often highly sensitive, and that privacy breaches are costly. To move towards stronger guarantees of privacy for synthetic medical data, we investigated the use of a differentially private training procedure for the GAN. Differential privacy is concerned with the influence of the presence or absence of individual records in a database. Intuitively, differential privacy places bounds on the probability of obtaining the same result (in our case, an instance of a trained GAN) given a small perturbation to the underlying dataset. If the training procedure guarantees $(\epsilon, \delta)$ differential privacy, then given two 'adjacent' datasets (differing in one record) $D$, $D'$,

$$P[\mathcal{M}(D) \in S] \leq e^\epsilon P[\mathcal{M}(D') \in S] + \delta \tag{1}$$

where $\mathcal{M}(D)$ is the GAN obtained from training on $D$, $S$ is any subset of possible outputs of the training procedure (any subset of possible GANs), and the probability $P$ takes into account the randomness in the procedure $\mathcal{M}(D)$. Thus, differential privacy requires that the distribution over GANs produced by $\mathcal{M}$ must vary 'slowly' as $D$ varies, where $\epsilon$ and $\delta$ bound this 'slowness'. Inspired by a recent preprint (Beaulieu-Jones et al., 2017), we apply the differential private stochastic gradient descent (DP-SGD) algorithm of (Abadi et al., 2016) to the discriminator (as the generator does not 'see' the private data directly). For further details on the algorithm (and the above definition of differential privacy), we refer to (Abadi et al., 2016) and (Dwork et al., 2006).

In practice, DP-SGD operates by clipping *per-example* gradients and adding noise in batches. This means the signal obtained from *any individual example* is limited, providing differential privacy. Some privacy budget is 'spent' every time the training procedure calculates gradients for the discriminator, which enables us to evaluate the effective values of $\epsilon$ and $\delta$ throughout training. We use the moments accountant method from (Abadi et al., 2016) to track this privacy spending. Finding hyperparameters which yield both acceptable privacy and realistic GAN samples proved challenging. We focused on the MNIST and eICU tasks with RCGAN, using the TSTR evaluation.

For MNIST, we clipped gradients to 0.05 and added Gaussian noise with mean zero and standard deviation $0.05 \times 2$. For $\epsilon = 1$ and $\delta \leq 1.8 \times 10^{-3}$, we achieved an accuracy of $0.75 \pm 0.03$. Sacrificing more privacy, with $\epsilon = 2$ and $\delta \leq 2.5 \times 10^{-4}$, the accuracy is $0.77 \pm 0.03$. These results are far below the performance reported by the non-private GAN (Table 1), highlighting the compounded difficulty of generating a realistic dataset while maintaining privacy. For comparison, in (Abadi et al., 2016) they report an accuracy of 0.95 for training an MNIST classifier (on the full task) on a real dataset in a differentially private manner. (Please note, however, that our GAN model had to solve the more challenging task of modeling digits as a time series.)

For eICU, the results are shown in Table 2b. For this case, we clipped gradients to 0.1 and added noise with standard deviation $0.1 \times 2$. In surprising contrast to our findings on MNIST, we observe that performance on the eICU tasks remains high with differentially private training. We fixed $\delta < 1/|D| (= 5.65 \times 10^{-5})$ (where $|D| = 17693$ is the number of training examples in eICU) for $\epsilon = 1$ by limiting to epochs before 64 (see Figure 5). While the best values for $\epsilon$ and $\delta$ ultimately depend on the use-case, setting $\delta < 1/|D|$ is a common heuristic Dwork et al. (2014).

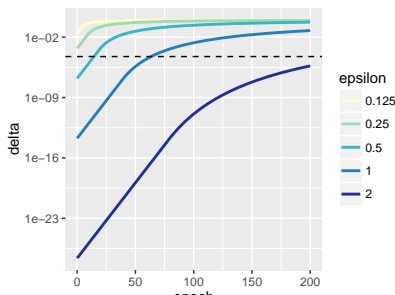

**Figure 5:** Probability of violating $\epsilon$-differential privacy ($\delta$) during training, for different values of $\epsilon$. Gaussian noise is added with mean zero and standard deviation $0.1 \times 2$. Dotted line shows $\delta = 1/|D|$, where $|D|$ is the number of training examples.

The true value of $\delta$ for the results in Table 2b is lower because the optimal epoch (assessed using the validation set) for most replicates was below 30.

Visual assessment of samples generated by the differentially-private GAN indicate that while it is prone to producing less-realistic sequences, the mistakes it introduces appear to be unimportant for the tasks we consider. In particular, the DP-GAN produces more extreme-valued sequences, but as the tasks are to predict extreme values, it may be that the most salient part of the sequence is preserved. The possibility to introduce privacy-preserving noise which nonetheless allows for the training of downstream models suggests interesting directions of research in the intersection of privacy and GANs.

## 8 CONCLUSION

We have described, trained and evaluated a recurrent GAN architecture for generating real-valued sequential data, which we call RGAN. We have additionally developed a conditional variant (RCGAN) to generate synthetic *datasets*, consisting of real-valued time-series data with associated labels. As this task poses new challenges, we have presented novel solutions to deal with evaluation and questions of privacy. By generating labelled training data - by conditioning on the labels and generating the corresponding samples, we can evaluate the quality of the model using the 'TSTR technique', where we train a model on the synthetic data, and evaluate it on a real, held-out test set. We have demonstrated this approach using 'serialised' multivariate MNIST, and on a dataset of real ICU patients, where models trained on the synthetic dataset achieved performance at times comparable to that of the real data. In domains such as medicine, where privacy concerns hinder the sharing of data, this implies that with refinement of these techniques, models could be developed on *synthetic* data that are still valuable for real tasks. This could enable the development of synthetic 'benchmarking' datasets for medicine (or other sensitive domains), of the kind which have enabled great progress in other areas. We have additionally illustrated that such a synthetic dataset does not pose a major privacy concern or constitute a data leak for the original sensitive training data, and that for stricter privacy guarantees, differential privacy can be used in training the RCGAN with some loss to performance.

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

Lantao Yu, Weinan Zhang, Jun Wang, and Yong Yu. SeqGAN: Sequence generative adversarial nets with policy gradient. 18 September 2016.

APPENDIX

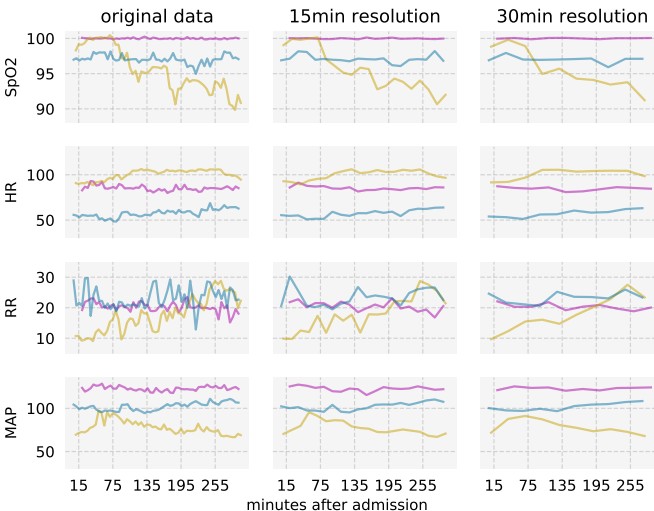

**Figure 6:** Data from three real eICU patients (purple, blue, gold) of the first five hours after admission. Noise from $\mathcal{N}(0, \sigma_e^2)$ for $\sigma_e = 0.1\sigma$ has been added to protect privacy, where $\sigma$ is the standard deviation of the true data (for that variable). We compare the data at its original sampling resolution with downsampled to one measurement every 15 minutes (the setting used in this paper) and 30 minutes. High-frequency fluctuations are lost through downsampling, but general trends and some variability are preserved in the 15 minute case. These patients were selected randomly from the set of patients with minimal missing data during the time period and so are representative of the cohort used to generate the training data.

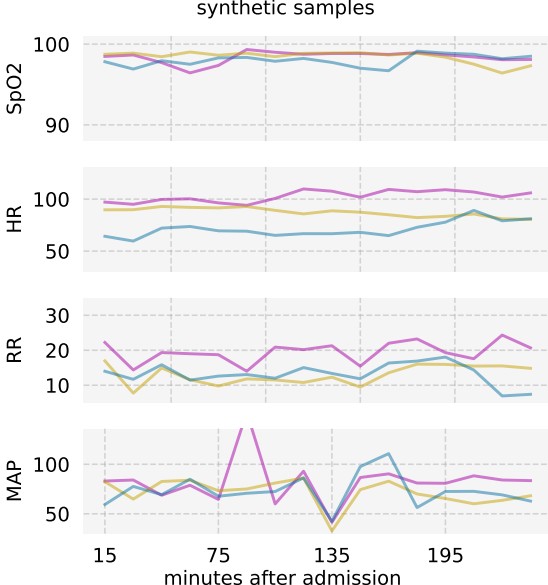

**Figure 7:** Three random samples from the generator trained on eICU data. These samples are from the synthetic datasets used in the TSTR experiments in section 5.1. The generator produces data in $[-1, 1]$, so to obtain medically relevant values, the inverse of the scaling transformation used on the training data has been applied. This transformation was to scale each variable at each time-point independently to the range $[-1, 1]$. An unusually high value (likely an artefact) in the mean arterial pressure at 135 minutes after admission is responsible for the apparent downward spike in the generated data.

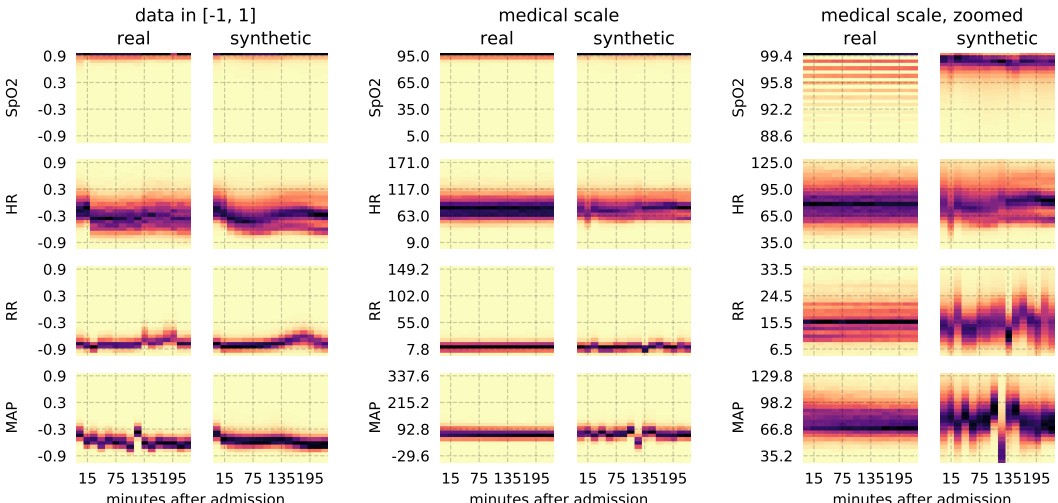

**(a)** Real and synthetic data in the range $[-1, 1]$. A scaling transformation $T$ is applied to the real data (independently for each variable at each timepoint), and the synthetic data is as produced by the generator.

**(b)** Real and synthetic data in medically meaningful units. For the real data, this is its original form. The synthetic data receives the inverse of the transformation $T$, producing some apparent discontinuities.

**(c)** Figure 8b zoomed into the regions of most variation. That SpO2 is integer-valued in the original data is now apparent.

**Figure 8:** Marginal distributions for each variable at each timepoint: Colour intensity shows the fraction of samples at that time point falling within a bin of values (25 bins over the full range) for that variable. Each subfigure shows the same data under different transformations (to and from $[-1, 1]$), and zooming in. The synthetic data consists of the generated datasets from all five replicates of the TSTR experiment in eICU described in section 5.1 with TSTR results reported in Table 2a. The real data is the training set for those expeirments.

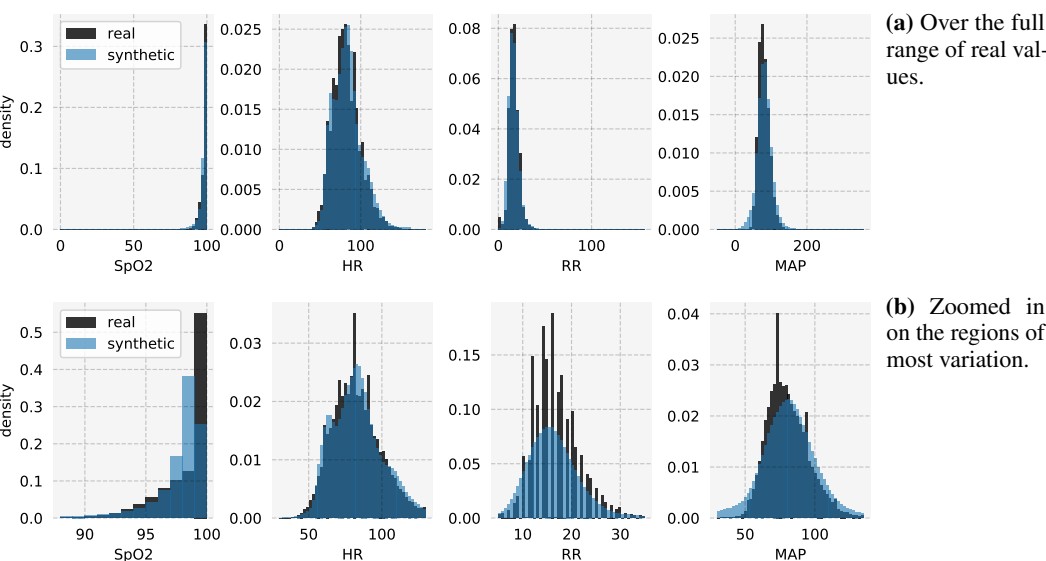

**(a)** Over the full range of real values.

**(b)** Zoomed in on the regions of most variation.

**Figure 9:** Histograms comparing the marginal distributions of each eICU variable (over all timepoints) between synthetic and real data. The synthetic data is that used in the TSTR experiments (five replicates) in section 5.1.

