# OpenReview forum: "Real-valued (Medical) Time Series Generation with Recurrent Conditional GANs"
_ICLR.cc/2018/Conference — Reject_

### Official Review · AnonReviewer1 · 2017-11-26
**interesting application but no clear takeaways**

**Rating:** 4
**Confidence:** 4

**Review:**

The authors propose to use synthetic data generated by GANs as a replacement for personally identifiable data in training ML models for privacy-sensitive applications such as medicine. In particular it demonstrates adversarial training of a recurrent generator for an ICU monitoring multidimensional time series, proposes to evaluate such models by the performance (on real data) of supervised classifiers trained on the synthetic data ("TSTR"), and empirically analyzes the privacy implications of training and using such a model.

This paper touches on many interesting issues -- deep/recurrent models of time series, privacy-respecting ML, adaptation from simulated to real-world domains. But it is somewhat unfocused and does not seem make a clear contribution to any of these.

The recurrent GAN architecture does not appear particularly novel --- the authors note that similar architectures have been used for discrete tasks such language modeling (and fail to note work that uses convolutional or recurrent generators for video prediction, a more relevant continuous task, see e.g.  http://carlvondrick.com/tinyvideo/, or autoregressive approaches to deep models of time series, e.g. WaveNet https://arxiv.org/abs/1609.03499) and there is no obvious new architectural innovation.

I also find it difficult to assess whether the proposed model is actually generating reasonable time series. It may be true that "one plot showing synthetic ICU data would not provide enough information to evaluate its actual similarity to the real data" because it could not rule out that case that the model has captured the marginal distribution in each dimension but not joint structure. However producing marginal distributions that look reasonable is at least a *necessary* condition and without seeing those plots it is hard to rule out that the model may be producing highly unrealistic samples.

The basic privacy paradigm proposed seems to be:
1. train a GAN using private data
2. generate new synthetic data, assume this data does not leak private information
3. train a supervised classifier on the private data
so that the GAN training-sampling loop basically functions as an anonymization procedure. For this to pan out, we'd need to see that the GAN samples are a) useful for a range of supervised tasks, and b) do not leak private information. But the results  in Table 2 show that the TSTR results are quite a lot worse than real data in most cases, and it's not obvious that the small set of tasks evaluated are representative of all tasks people might care about. The attempts to demonstrate empirically that the GAN does not memorize training data aren't particularly convincing; this is an adversarial setting so the fact that a *particular* test doesn't reveal private data doesn't imply that a determined attacker wouldn't succeed. In this vein, the experiments with DP-SGD are more interesting, although a more direct comparison would be helpful (it is frustrating to flip back and forth between Tables 2 and 3 in an attempt to tease out relative performance) and and it is not clear how the settings (ε = 0.5 and δ ≤ 9.8 × 10−3) were selected or whether they provide a useful level of privacy. That said I agree this is an interesting avenue for future work.

Finally it's worth noting that discarding patients with missing data is unlikely to be innocuous for ICU applications; data are quite often not missing at random (e.g., a patient going into a seizure may dislocate a sensor). It appears that the analysis in this paper threw out more than 90% of the patients in their original dataset, which would present serious concerns in using the resulting synthetic data to represent the population at large. One could imagine coding missing data in various ways (e.g. asking the generator to produce a missingness pattern as well as a time series and allowing the discriminator to access only the masked time series, or explicitly building a latent variable model) and some sort of principled approach to missing data seems crucial for meaningful results on this application.

---

> ### Author Response · Authors · 2018-01-02
> **comments and updates for AnonReviewer1**
>
> Thank you for your feedback and comments.
>
> Regarding your concern about the lack of clear contributions in this work, we would like to clarify that the work is ultimately focused on generating synthetic medical time series data. Achieving this necessitated multiple elements - finding and developing an appropriate architecture and making it work in this domain, finding and developing evaluation techniques for synthetic time series, analysing the privacy implications and testing a differential private training algorithm in combination with RGANs. Any one of these topics could constitute an independent research project, and we have necessarily not addressed each to the fullest extent possible, but it was important to address all components in the pursuit of our objective. Without an evaluation measure, we cannot judge whether the method really works beyond visual assessment. Also with medical data the privacy question immediately pops up and earlier versions of this manuscript were rejected because we didn’t cover differential privacy. It therefore appears necessary to include all the pieces into one manuscript to have a coherent piece of work. We appreciate the pointers to other recurrent generators and have extended the related work section accordingly.
>
> Following your suggestion, we have added several figures to an appendix, including comparisons of marginal distributions between the synthetic and real eICU data. We compare both the marginal distributions of each variable at each time point (figure 8) and histograms of each variable ignoring the temporal component (figure 9). As you have already noted, such marginal distributions are still imperfect, but at least by this measure the synthetic and real distributions are roughly similar. The most obvious differences arise from the real data being integer-valued (in the case of SpO2) and our method of scaling the synthetic data back into medical ranges. We thank you for proposing to include these plots, since they are very helpful as a sanity check, and highlighted in particular the weakness in our choice of data scaling during training (now discussed in the appendix).
>
> We elected to focus on ICU tasks pertaining to early warning of extreme values because these simplified endpoints can contribute to more sophisticated early warning systems, which we work on in collaboration with clinicians in other projects. Developing a realistic, clinically useful system for intensive care is beyond the scope of this work.
>
> Regarding the privacy analysis, we agree that the tests we have done are not all-encompassing, however we intentionally focused on the question of whether or not the generator preferentially generates samples from the training set (assessed with established methods using maximum mean discrepancy), and not whether the samples are robust against any conceivable attack. We felt this question of memorisation would be of broader interest to the GAN field, while allowing us to make some weaker claims about the privacy of the original data.
>
> Regarding the TSTR results in the differentially private setting: to make comparison easier, we have merged tables 3 and table 2 into two sub-tables. We have updated the results in Table 3 (now Table 2b) in light of choosing of less arbitrary cut-off for the acceptable delta value. Following a common heuristic (cited in the revised manuscript) we require delta to be below 1/|D| where |D| is the size of the training set, and only consider epochs satisfying this criterion (we show how delta increases for different epsilon values during training in a new Figure 5 in the main body of the paper). This caused minor changes in the TSTR results.
>
> We are conscious of the fact that discarding patients with missing data may limit the generalisability of the model. In this case, the majority of excluded patients were missing measurements of mean arterial pressure (MAP), which is an invasive measurement. Thus, any model built on this data is restricted to patients with measurements of MAP. However, since those patients with MAP measurements are typically more critical, building an early warning system restricted to such patients is not unreasonable. Also, we don’t claim and in this work we don’t aim that our approach could or should be used on patients outside the cohort of interest. Nevertheless, restricting to cohorts is common practice in medical science.
>
> Regarding accounting for missing data, this is a very interesting direction for this research, and we are looking into it. For the use-case in ML for healthcare, building realistic missingness patterns is particularly relevant. However, for methods developments it is important to separate the different challenges and to tackle one problem at a time. For this work we considered the problem of generating medical time-series without (much) missing data. Medical data harbors many more challenges that needs answers before such a system can be used for the benefit of patients.

---

### Official Review · AnonReviewer2 · 2017-11-27
**This paper is a good extension of GANs and generative RNNs to the continuous domain time series setting**

**Rating:** 6
**Confidence:** 4

**Review:**

In this paper, the authors propose a recurrent GAN architecture that generates continuous domain sequences. To accomplish this, they use a generator LSTM that takes in a sequence of random noise as well as a sequence of conditonal information and outputs a sequence. The discriminator LSTM takes a sequence (and conditional information) as input and classifies each element of the sequence as real or synthetic -- the entire sequence is then classified by vote. The authors evaluate on several synthetic tasks, as well as an ICU timeseries data task.

Overall, I thought the paper was clearly written and extremely easy to follow. To the best of my knowledge, the method proposed by the authors is novel, and differs from traditional sentence generation (as an example) models because it is intended to produce continuous domain outputs. Furthermore, the story of generating medical training data for public release is an interesting use case for a model like this, particularly since training on synthetic data appears to achieve not competitive but quite reasonable accuracy, even when the model is trained in a differentially private fashion.

My most important piece of feedback is that I think it would be useful to include a few examples of the eICU time series data, both real and synthetic. This might give a better sense of: (1) how difficult the task is, (2) how much variation there is in the real data from patient to patient, and (3) how much variation we see in the synthetic time series. Are the synthetic time series clearly multimodal, or do they display some of the mode collapse behavior occasionally seen in GANs?

I would additionally like to see a few examples of the time series data at both the 5 minute granularity and the 15 minute granularity. You claim that downsampling the data to 15 minute time steps still captures the relevant dynamics of the data -- is it obvious from the data that variations in the measured variables are not significant over a 5 minute interval? As it stands, this is somewhat an unknown, and should be easy enough to demonstrate.

---

> ### Author Response · Authors · 2018-01-02
> **comments and updates for AnonReviewer2**
>
> Thank you for your feedback. Following your suggestions and those from other reviewers, we have extended the manuscript, including several new figures in an appendix.
>
> As suggested, in Figure 6 we now show samples from three random real eICU patients, both at their original sampling resolution (approximately 5 minute granularity) and downsampled to 15 minute and 30 minute granularity. The downsampling loses some variation (high frequency fluctuations naturally), but over the 4-5 hour period of interest, at 15 minute granularity the main trends and variations are still visible.
>
> In Figure 7 we show three random synthetic patients. Regarding mode collapse, based purely on Figures 6 and 7 there is some evidence that the GAN produces SpO2 traces that hover near 99/100 and fluctuate, whereas in the real data patients sometimes stay perfectly at 100, and others degrade more steadily.  If the GAN is failing to capture ‘degrading modes’ of patients, we would expect low TSTR scores in the early warning tasks, which is partially observed, but could be explained by other aspects of the synthetic data. We observed some mode collapse of the RCGAN in producing MNIST digits (more readily observable), so we don’t think this architecture is intrinsically resistant to it. However, the eICU data doesn’t exhibit any obviously multimodal behaviour.
>
> To further study the general properties of the synthetic eICU data, we also produced a set of marginal distributions in Figures 8 and 9, providing further evidence that the synthetic data has captured the main characteristics of the real data.
>
> These additional figures will give some additional information about the data that may help the reader, but we consider a developing a specifically tuned model for the eICU being beyond the scope of this data.

---

### Official Review · AnonReviewer3 · 2017-12-04
**an interesting approach and generates time series sequences, however the medical use case needs work.**

**Rating:** 5
**Confidence:** 4

**Review:**

This paper proposes to use RGANs and RCGANS to generate synthetic sequences of actual data. They demonstrate the quality of the sequences on sine waves, MNIST, and ICU telemetry data.

The authors demonstrate novel approaches for generating real-valued sequences using adversarial training, a train on synthetic, test of real and vice versa method for evaluating GANS, generating synthetic medical time series data, and an empirical privacy analysis.

Major
- the medical use case is not motivating. de-identifying the 4 telemetry measures is extremely easy and there is little evidence to show that it is even possible to reidentify individuals using these 4 measures. our institutional review board would certainly allow self-certification of the data (i.e. removing the patient identifiers and publishing the first 4 hours of sequences).
- the labels selected by the authors for the icu example are to forecast the next 15 minutes and whether a critical value is reached. Please add information about how this critical value was generated. Also it would be very useful to say that a physician was consulted and that the critical values were "clinically" useful.
- the changes in performance of TSTR are large enough that I would have difficulty trusting any experiments using the synthetic data. If I optimized a method using this synthetic data, I would still need to assess the result on real data.
- In addition it is unclear whether this synthetic process would actually generate results that are clinically useful. The authors certainly make a convincing statement about the internal validity of the method. An externally valid measure would strengthen the results. I'm not quite sure how the authors could externally validate the synthetic data as this would also require generating synthetic outcome measures. I think it would be possible for the synthetic sequence to also generate an outcome measure (i.e. death) based on the first 4 hours of stay.

Minor
- write in the description for table 1 what task the accuracies correspond.

Summary
The authors present methods for generating synthetic sequences. The MNIST example is compelling. However the ICU example has some pitfalls which need to be addressed.

---

> ### Author Response · Authors · 2018-01-02
> **comments and updates for AnonReviewer3**
>
> Thank you for your comments and suggestions.
>
> - We agree that de-identifying these specific variables is not difficult, and unlikely to pose issues for data sharing. However, the approach we have taken in this work is to assume that the data is private (for whatever reason, IRB compliance is one example) and to work from there. We think this is a reasonable approach because the difficulty of data-sharing is a common complaint in machine learning applied to healthcare, and having methods to enable data-sharing independent of the specifics of the data addresses that. It also allows us to avoid answering the question of which specific aspects of ICU time-series need protecting. The answer to that question lies somewhere between ‘nothing’ and ‘everything’, depending on the temporal resolution, the length of the time series, and the number of variables released, the laws of each specific country, and is arguably a full research question on its own. Hence, we decided to focus on the idealised case of commonly-measured variables.
>
> - We agree that for several of the tasks, the TSTR performance with synthetic data is noticeably worse, indicating that the synthetic data isn't capturing all properties of the real data. While the performance may still be acceptable in some settings (for example, where false discoveries are less harmful), the purpose of the evaluation is not to demonstrate that the data is optimal for these specific tasks, but that the data is sufficiently realistic generally speaking. Of course, if real data is available it should be preferred, but in the case where this is not possible, we show that even the reduced performance from synthetic data can be useful.
>
> - The objective of the TSTR method is to assess how useful the synthetic data is. Here, we focused on an early-warning system situation, which is a very relevant task in intensive care medicine. An interesting external validation would be to use the classifier trained on the eICU-derived synthetic data, and to test it on another ICU dataset, such as MIMIC-III, to test cross-hospital generalisation. We will investigate this for the next revision of the manuscript.
>
> - Demonstrating the system in a realistic, clinically useful setting is not the intention of this work. As it is common in the ML field, we choose datasets to demonstrate the capabilities of the new methodologies, without demonstrating the full use in practice. That was done so for the MNIST dataset in this paper and any papers in the past, without demonstrating that the new classifier indeed would make a difference in practice. Implementing such a system goes beyond what can typically be done in a conference paper. Similarly, to illustrate usefulness, we would need to train the system on a larger set of identifiable variables, make sure that it predicts something clinically relevant (which requires involvements of physicians) and then convince an IRB that the generated data by our system does not leak private data. However, this work is about the technical basis to perform such work in the future. We therefore consider the demonstration of clinical usefulness of the system beyond the scope of this work.
>
> - Clarification: the tasks pertain to the patient’s values in the next hour, not 15 minutes. We have added details in the revised manuscript on how the critical thresholds were obtained. Briefly, we looked at the distribution of the data, so no clinician was needed. We also cross-referenced with easily-obtained healthy ranges to make sure our ICU population didn’t deviate too strongly from the norm.
>
> - Minor: we have extended the description of Table 1 as requested.

---

### Decision · Program_Chairs · 2018-01-29
**ICLR 2018 Conference Acceptance Decision**

**Decision:**

Reject

**Comment:**

Overall I agree with the assessment of R1 that the paper touches on many interesting issues (deep learning for time series, privacy-respecting ML, simulated-to-real-world adaptation) but does not make a strong contribution to any of these. Especially with respect to the privacy-respecting aspect, there needs to be more analysis showing that the generative procedure does not leak private information (noting R1 and R3’s comments). I appreciate the authors clarifying the focus of the work, and revising the manuscript to respond to the reviews. Overall it’s a good paper on an important topic but I think there are too many issues outstanding for accept at this point.